# Regulation of Histone Ubiquitination in Response to DNA Double Strand Breaks

**DOI:** 10.3390/cells9071699

**Published:** 2020-07-16

**Authors:** Lanni Aquila, Boyko S. Atanassov

**Affiliations:** Department of Pharmacology and Therapeutics, Roswell Park Comprehensive Cancer Center, Buffalo, NY 14263, USA; Lanni.Aquila@RoswellPark.org

**Keywords:** DNA damage response, double strand break repair, DSBs, histones, ubiquitination, deubiquitinases, deubiquitinating enzymes, DUBs

## Abstract

Eukaryotic cells are constantly exposed to both endogenous and exogenous stressors that promote the induction of DNA damage. Of this damage, double strand breaks (DSBs) are the most lethal and must be efficiently repaired in order to maintain genomic integrity. Repair of DSBs occurs primarily through one of two major pathways: non-homologous end joining (NHEJ) or homologous recombination (HR). The choice between these pathways is in part regulated by histone post-translational modifications (PTMs) including ubiquitination. Ubiquitinated histones not only influence transcription and chromatin architecture at sites neighboring DSBs but serve as critical recruitment platforms for repair machinery as well. The reversal of these modifications by deubiquitinating enzymes (DUBs) is increasingly being recognized in a number of cellular processes including DSB repair. In this context, DUBs ensure proper levels of ubiquitin, regulate recruitment of downstream effectors, dictate repair pathway choice, and facilitate appropriate termination of the repair response. This review outlines the current understanding of histone ubiquitination in response to DSBs, followed by a comprehensive overview of the DUBs that catalyze the removal of these marks.

## 1. Introduction

Double strand breaks (DSBs), the most deleterious form of damage endured by eukaryotic cells, can be induced by exogenous exposures including ionizing radiation (IR) and radiomimetic drugs, or endogenously via replication fork collapse. DSBs can also be programmed in order to facilitate genetic diversity during meiosis or during variable, diversity, and joining (V(D)J) recombination and antibody class switching of developing lymphocytes. If left unrepaired, DSBs contribute to the accumulation of chromosomal aberrations and translocations that can ultimately promote tumorigenesis. The presence of a DSB triggers changes in transcriptional programming and induces cell cycle arrest to allow for proper repair. Two major repair pathways have evolved for DSBs: non-homologous end joining (NHEJ) and homologous recombination (HR), which have both been extensively reviewed [1,2,3,4]. In short, NHEJ, which can be activated throughout the cell cycle, involves the direct ligation of broken ends. Due to minimal end processing, NHEJ is generally thought to be error-prone. HR, on the other hand, is restricted to late S and G2 phases of the cell cycle due to the requirement for a homologous repair template. HR involves resecting the ends surrounding a DSB to allow for invasion of the template and proper, error-free repair. These repair pathways are precisely orchestrated by complex signaling networks that involve damage sensors, signal transducers, and repair effectors.

Importantly, DSBs occur in the context of chromatin and can therefore influence and be influenced by histone post-translational modifications (PTMs). PTMs, including phosphorylation, methylation, and ubiquitination, amongst others, occur on the tails of core histones H2A, H2B, H3, and H4, as well as on linker histone H1. Ubiquitinated histones, which will be the focus of this review, can not only alter chromatin structure before and after repair, but can propagate damage signals and act as recruitment platforms for repair machinery as well.

Ubiquitin is a highly conserved 76 amino acid protein that can be covalently attached to lysine residues on histone and non-histone target proteins. Ubiquitination occurs through an enzymatic cascade involving E1 activating enzymes, E2 conjugating enzymes, and E3 ubiquitin ligases. Addition of a single ubiquitin moiety to a target protein is termed monoubiquitination. Monoubiquitination at additional lysine residues of the same target protein is referred to as multi-monoubiquitination, while subsequent rounds of this ubiquitination cascade at the same lysine leads to the formation of polyubiquitin chains. Polyubiquitin chains are formed as linkages via the N-terminal amine or one of seven lysine residues within the ubiquitin molecule itself: lysine 6 (K6), K11, K27, K29, K33, K48, and K63. Linear polyubiquitin chains consist of one linkage type, while mixed chains contain multiple linkage types. Polyubiquitin chains can also be branched, where the ubiquitin molecule itself is ubiquitinated at more than one lysine. Ubiquitination, specifically K48-linked polyubiquitin, was first recognized for its ability to target substrates for proteasomal degradation. It is now appreciated that ubiquitin regulates many aspects of a substrate’s fate including localization, activity, and protein–protein interactions. While histone ubiquitination was first identified as a regulator of transcription, it is now appreciated that these PTMs play multifaceted roles in the context of DSB repair.

Ubiquitination is a dynamic PTM that is reversible through the action of deubiquitinating enzymes (DUBs). As their name suggests, these enzymes primarily function to remove or edit ubiquitin chains. The mammalian genome encodes over 100 DUBs which have been classified into five major families: 1) Jab1/MPN domain-associated metalloisopeptidases (JAMMs); 2) ubiquitin specific proteases (USPs), which make up the largest family of DUBs; 3) ubiquitin C-terminal hydrolases (UCHs); 4) Machado-Josephin domain proteases (MJDs); and 5) ovarian tumor proteases (OTUs). The JAMM family of DUBs are unique in that they are metalloproteases, requiring coordination of zinc for catalytic activity. The remainder of DUBs are cysteine proteases, characterized by a catalytic cysteine residue within their active site. Two additional families of cysteine proteases have recently been characterized: motif interacting with ubiquitin-containing novel DUB family (MINDYs) and zinc finger and UFSP domain proteins (ZUFSPs) [5,6]. DUBs and the consequences of deubiquitination have been far less studied but are now being implicated in virtually every cellular process. Histone deubiquitination has specifically been implicated in transcriptional regulation [7,8], stem cell maintenance [9], and DNA damage and repair.

This review will outline the current understanding of how histone ubiquitination contributes to the successful repair of DSBs as well as highlight the DUBs that directly target these modifications to regulate this response.

## 2. Histone Ubiquitination as a Response to DSBs

### 2.1. The RNF8-RNF168 Axis

DSBs are characterized by and often observed in the laboratory through the accumulation of repair machinery at distinct foci known as ionizing radiation-induced foci (IRIF). These foci are formed following a series of phosphorylation and ubiquitination events, beginning with the activation of sensor kinases, most notably ataxia-telangiectasia mutated (ATM). A major target of ATM phosphorylation is Serine 139 of H2AX, an H2A variant. Commonly referred to as γH2AX, this hallmark of DSBs extends up to 2 Mb around sites of damage and acts as the key initiating event for downstream signaling [10,11]. γH2AX recruits MDC1 which is subsequently phosphorylated by ATM, leading to recruitment of the E3 ligase RNF8 [12,13]. RNF8 was originally thought to interact with and polyubiquitinate H2A and H2AX to recruit additional components of the repair machinery [14,15,16,17]. However, RNF8 alone is insufficient in sustaining ubiquitination at DSB sites and was later deemed a recruitment platform for a second E3 ligase, RNF168 [18,19,20]. More recent studies were unable to detect RNF8 activity towards H2A-type histones and suggest that RNF168 is responsible for stabilizing and propagating the ubiquitination signal [21]. RNF8 has since been proposed to catalyze K63-polyubiquitination on linker histone H1 or other non-histone proteins including polycomb group protein L3MBTL2 to facilitate RNF168 recruitment [21,22,23,24]. These models are not mutually exclusive. RNF8 may also have activity towards H2B [25] and may be able to reciprocally extend RNF168-mediated chains, suggesting potential crosstalk between these ubiquitin modifications [21]. Interestingly, RNF8 was found to auto-ubiquitinate itself with K48-linked polyubiquitin chains [26]. Polyubiquitinated RNF8 is subsequently processed by p97 (also known as VCP or CDC48), a segregase that uses energy from ATP to remove ubiquitinated substrates from their cellular location [26]. In an effort to identify regulators of RNF8 degradation, this same group identified Ataxin 3 (ATXN3), a p97-associated DUB that has the ability to cleave not only K48-linked polyubiquitin chains, but K63-linked and mixed polyubiquitin chains as well [27,28]. Enhanced RNF8 ubiquitination and rates of degradation were found upon depletion of p97 or ATXN3 [26]. This same group investigated the role of the p97-ATXN3-RNF8-proteasome network in the response to DNA damage, echoing work done in *Caenorhabditis elegans* by Ackermann et al. [29]. Both p97 and ATXN3 are localized to sites of DSBs. p97 or ATXN3 depleted cells have increased levels of DSBs and are hypersensitive to IR. These defects can be rescued with ectopic expression of wild type (WT), but not catalytically inactive or ubiquitin binding-defective ATXN3. Depletion of p97 or ATXN3 led to hyperaccumulation of RNF8 and K63-linked polyubiquitin chains at DSBs, although this is at odds with a recent publication that observed reduced levels of RNF8 at sites of damage in depleted cells [26,30]. Despite these findings, levels of ubiquitinated H1 (a mark catalyzed by RNF8) and recruitment of downstream effectors RNF168 and 53BP1 were reduced, suggesting unbalanced RNF8 and K63-linked polyubiquitin chains impair the DSB response. Ultimately, these data suggest that p97 and ATXN3 regulate RNF8 homeostasis in order to prevent its hyperaccumulation at DSB sites and thus promote successful NHEJ. In support of this, depletion of ATXN3 led to reduced NHEJ efficiency and increased end resection, a hallmark of HR repair [26]. The recruitment and activity of RNF8 and RNF168 is rather complex and is not dependent on a single factor. It is generally accepted that following RNF8-mediated ubiquitination events, RNF168 preferentially catalyzes monoubiquitination or K63-linked polyubiquitination of lysines 13 and 15 of H2A (H2AK13/15) and can in turn recognize ubiquitinated H2A in order to further transmit this signal [21,31,32,33,34]. Interestingly, RNF168 has also been shown to assemble non-canonical K27-linked polyubiquitin chains on H2A [31]. Ultimately, the coordinated activities of RNF8 and RNF168 form docking sites for downstream factors in the DSB response, most notably RAP80 and 53BP1 (Figure 1A).

Through ubiquitin interacting motifs (UIMs) and a sumoylation interacting motif (SIM), RAP80 is recruited to K63-linked polyubiquitin chains and sumoylation conjugates at damage sites, respectively [35,36,37]. SUMO, a small ubiquitin-like PTM, has been detected at sites of DSBs. RNF4, a SUMO-targeted ubiquitin E3 ligase, catalyzes the ubiquitination of SUMO moieties present on target proteins, resulting in the formation of SUMO-ubiquitin hybrid chains. Interestingly, RAP80 recognizes these hybrid chains with an affinity ~80-fold greater than that for SUMO or K63-linked polyubiquitin chains alone [37]. Both UIM and SIM domains of RAP80 have been found to be required for proper RAP80 recruitment to DSB sites [36,37], which again illustrates the crosstalk between various PTMs in the response to DSBs. Ultimately, RAP80 recruitment to DSBs facilitates assembly of the BRCA1-A complex, which consists of BRCA1, Abraxas, MERIT40, and deubiquitinating enzymes, BRCC45 and BRCC36. Briefly, as this complex will be discussed in more detail below, BRCA1-A prevents hyperactivation of HR by restricting end resection [38,39,40]. 53BP1 is a second major player recruited to sites of DSBs. This occurs through dual recognition of RNF8-mediated H2AK15ub as well as H4K20 dimethylation (H4K20me2) (facilitated by a ubiquitin recognition motif and TUDOR domain respectively) [34,41]. H4K20me2 is also bound by other TUDOR domain-containing proteins including JMJD2A (see below) and L3MBTL1. Upon DNA damage, L3MBTL1 is sent for proteasomal degradation following its RNF8-mediated K48-linked polyubiquitination. The removal of L3MBTL1 from chromatin exposes H4K20me2 and enables 53BP1 binding [41]. Interestingly, p97 can be directed to RNF8-deposited K48-linked polyubiquitin chains via its adaptor proteins UFD1 and NPL4 [42,43,44,45]. Depletion of p97, UFD1, NPL4 or expression of a segregase mutant p97 results in decreased DSB repair efficiency and hypersensitivity to IR. Accordingly, these defects are accompanied by increased levels of K48-polyubiquitin chains at sites of DNA damage [43,44]. The current model suggests that the combination of polyubiquitinated L3MBTL1 and p97 segregase activity leads to the extraction of L3MBTL1 from chromatin to allow for proper 53BP1 assembly at H4K20me2 marks [45]. In support of this, displacement of L3MBTL1 from DSB sites is impaired when p97 is inactive, resulting in reduced 53BP1 accumulation at IRIF [43,45]. Ultimately, once recruited to sites of DSBs, 53BP1 promotes NHEJ through the recruitment of anti-end resection factors and suppression of BRCA1 [46,47,48,49].

### 2.2. H2AK119 Monoubiquitination Links Transcriptional Regulation to the DSB Response

H2A was the first ubiquitinated protein to be identified and comprises between 5% and 15% of total cellular H2A [50]. The monoubiquitinated residue was first mapped to K119 (H2AK119ub1) and was initially speculated to be involved in transcriptional regulation [51,52]. The polycomb repressive complex 1 (PRC1) has been identified as a major regulator of this mark, solidifying a role for H2A monoubiquitination in gene repression. Specifically, an E3 ligase (RING1A/RNF2) along with a polycomb-group protein (PcG) subunit, such as BMI1, make up the core heterodimer of PRC1 that catalyzes H2AK119ub1. Classically, H2AK119ub1 has been implicated in X-inactivation and polycomb-mediated transcriptional silencing [53,54,55,56]. The initial observation that RNF2 and BMI1 relocate from polycomb bodies [57,58] to IRIF suggested that transcriptional silencing takes place in the context of DSBs as well (Figure 1B). This finding expanded on the first report connecting transcription to DSBs by Kruhlak et al., detailing a decrease in RNA polymerase I transcription in irradiated cells [59]. BMI1 foci, in addition to several other PRC1 subunits, were found to colocalize with γH2AX, ATM, and MDC1 [60,61,62,63]. Depletion of either RNF2 or BMI1 enhances sensitivity to IR and decreases HR efficacy, further suggesting a role for PRC1 E3 ligase activity in the DSB response [60,62]. BMI1 is required for DSB-induced H2AK119ub1 and these foci colocalize with 53BP1 [64]. Utilizing a novel reporter construct, the Greenberg lab was able to simultaneously visualize DSB repair and measure local levels of transcription [65]. Increased ubiquitination of H2A was observed upon induction of DSBs, followed by a corresponding decrease in transcription up to kilobases away from the site of damage [64,65]. While total RNA polymerase II expression was unchanged and therefore could not explain this decrease in transcription, the levels of RNA polymerase II phosphorylation at serine 2, indicative of an elongating polymerase, were decreased [65]. In line with this finding, H2AK119ub1 was originally suggested to repress transcription through preventing elongation [66,67]. Further confirming a role for H2AK119ub1, levels of transcription were restored upon BMI1 depletion or expression of H2AK119/120R, an H2A mutant that is unable to be ubiquitinated at lysines 119 and 120 [64,65]. There is not yet a consensus on the precise mechanism directing PcGs to sites of DSBs, and a more recent report was unable to detect PRC1 at sites of restriction enzyme-induced DSBs entirely, warranting future investigations [61,68,69,70,71]. RNF2 and BMI1 have also been identified as interacting partners of H2AX, and this association is enhanced in irradiated cells. While monoubiquitinated H2AX is rapidly induced upon IR, knockdown of either RNF2 or BMI1 impaired accumulation of this mark [63,71,72]. Using H2AX mutants, RNF2/BMI1 was specifically shown to ubiquitinate H2AX at K119/120 and the expression of this mutant increases sensitivity to IR [63,72]. It was further identified that loss of RNF2/BMI1-mediated ubiquitination at H2AXK120 may impair γH2AX and phosphorylated ATM foci, as well as the accumulation of downstream mediators of the DSB response including MDC1, BRCA1, and 53BP1 [60,63]. This is another example of PTM crosstalk and suggests that H2A monoubiquitination may also serve to orchestrate and maintain IRIF.

### 2.3. BRCA1-BARD1 Regulation of Repair Pathway Choice Via H2AK125/127/129 Ubiquitination

The best characterized interacting partner of BRCA1 is the BRCA1-associated RING domain protein 1 (BARD1). BARD1 and BRCA1 interact through their N-terminal RING domains and together form an active E3 ubiquitin ligase [73,74,75]. Early reports suggested that BRCA1-BARD1 is critical for repair of chromosomal breaks [76] and in vitro work demonstrated that BRCA1-BARD1 catalyzes K6-linked polyubiquitin chains [77,78], although the function of these polyubiquitin linkages in DSB repair have yet to be elucidated. While the role of BRCA1-BARD1 in DSB repair has been controversial [79], the heterodimer has recently been shown to colocalize with ubiquitin conjugates following DNA damage. BRCA1-BARD1 specifically catalyzes ubiquitination of nucleosomal H2A at K125/127/129 [80]. Work by the Morris lab has further characterized the role of BRCA1-BARD1 in DSB repair by demonstrating that siRNA-mediated knockdown of BARD1 led to enhanced sensitivity to DNA damaging agents. Sensitivity was restored upon expression of WT BARD1, but not the L44R mutant that is unable to heterodimerize and does not have E3 ligase activity or the R99E mutant that can heterodimerize but is also E3 ligase defective [81]. This suggested a role for BRCA1-BARD1 E3 ligase activity in the promotion of HR and prompted the further investigation of the role of this heterodimer in DSB repair pathway choice, which ultimately depends on the extent of DNA end resection. BARD1-L44R or R99E-expressing cells exhibited fewer, smaller, and less intense foci of end resection factors RPA and RAD51 when compared to WT BARD1-expressing cells, indicating that the heterodimer functions to promote end resection [81]. 53BP1, which inhibits end resection, is normally relocalized to the periphery of IRIF upon BRCA1 recruitment to facilitate end resection. Reduced peripheral 53BP1 was observed in cells lacking BRCA1-BARD1 E3 ligase activity. Furthermore, cells lacking BRCA1-BARD1 E3 ligase activity had shorter resection lengths and it was ultimately determined that this heterodimer promotes end resection after the cell has committed to HR mediated repair [81]. In this context, H2A was identified as a relevant BRCA1-BARD1 substrate. Interestingly, expression and incorporation of an H2A-ubiquitin fusion into chromatin restored RAD51 foci and sensitivity to DNA damaging agents in a dose-dependent manner [81]. To further understand the role of BRCA1-BARD1 and H2A ubiquitination in the DSB response, this group investigated the role of SMARCAD1 (Fun30 in yeast), a promoter of DNA end resection. SMARCAD1 has two ubiquitin binding domains and directly interacts with ubiquitinated H2A. SMARCAD1 is localized to sites of DSBs and this localization is reduced when BARD1 is depleted or when SMARCAD1 ubiquitin binding domains are mutated. A model has been proposed in which BRCA1-BARD1 ubiquitinates H2AK125/127/129 in response to DSB induction (Figure 1C). This enhances the recruitment of SMARCAD1 to sites of damage to either reposition or evict nucleosomes, effectively removing 53BP1 from the core of IRIF. Ultimately, these events promote end resection and the successful completion of HR-mediated repair [81]. 

As mentioned above, BRCA1 is also recruited to DSB sites as a component of the BRCA1-A complex in order to restrict end resection. It is important to note that BRCA1 is a part of at least three other complexes that have been implicated in DSB repair [39]. The BRCA1-B complex, comprised of BRCA1, TopBP1, and BACH1, has been shown to be required for S phase progression [82] and cell cycle checkpoints following DNA damage [83]. BRCA1 in association with CtIP and MRN form the BRCA1-C complex that acts to promote end resection in order for successful HR. Although the above-mentioned work focuses on the BRCA1-BARD1 heterodimer for its E3 ligase activity, the formation of these distinct complexes suggests that BRCA1 plays varying roles in the DSB response. This may be due to different timing, localization, or cell cycle dependent complex formation. For example, as mentioned above, the BRCA1-A complex is directed to sites of damage following a signaling cascade after the phosphorylation of H2AX. BRCA1-B and BRCA1-C complexes, however, have been shown to be recruited to DSBs independently of γH2AX. These complexes, in addition to BARD1, are recruited to DSBs through PARylation modifications which reportedly appear at DSB sites much earlier than γH2AX [84]. This points to temporal regulation of these complexes. 

### 2.4. A Non-Canonical Role of H2B Ubiquitination

While H2AK119ub1 has been mostly implicated in transcriptional silencing, monoubiquitination of H2B promotes transcriptional elongation and activation [85,86,87]. In mammals, this monoubiquitination has been mapped to K120 (H2BK120ub1) and is primarily catalyzed by the RNF20-RNF40 heterodimer (Figure 1D) [88,89]. H2BK120ub1 levels increase following exposure to IR or treatment with radiomimetics, and although direct detection of RNF20-RNF40 at IRIF has been debated, this increase in ubiquitination is reversed upon RNF20 depletion [90,91,92]. Of note, treatment with DSB-inducing agents increases global levels of H2BK120ub1, but not other histone modifications associated with the activation of transcription such as H3K4me3. Likewise, depletion of RNF20-RNF40 does not influence other transcription-associated PTMs [92]. H2BK120ub1 levels remain elevated following damage even in the presence of a transcription inhibitor, further suggesting a specific role for H2BK120ub1 in the DSB response [91]. Depletion of RNF20-RNF40 led to less efficient NHEJ and HR as assessed by GFP reporter constructs. A later report noted loss of this heterodimer contributes to defects in class switch recombination (CSR), a process by which NHEJ mediates repair of programmed DSBs to facilitate antibody isotype switching, further implicating a role for H2BK120ub1 in NHEJ [91,92]. RNF20-RNF40-mediated H2BK120ub1 has also been implicated in regulating accumulation of XRRC4 and Ku80, major mediators of NHEJ, as well as end resection factors including RPA, RAD51, and BRCA1 [90,91]. Considering the established localization of BRCA1 to ubiquitinated H2A(X), this data again suggests there are multiple pathways at play that may differentially regulate distinct BRCA1 complexes. This phenomenon may additionally represent early and late responses to DSBs, characterized by ubiquitinated H2A(X) and H2B, respectively. Reports that depletion of RNF20 does not influence phosphorylation and ubiquitination of H2A(X) and vice versa further support this claim [90,92], although another group observed decreased clearance of γH2AX and 53BP1 upon depletion of RNF20-RNF40 [91]. Interestingly, RAP80, a prominent recruiter of BRCA1 to DSB foci, was shown to interact with ubiquitinated H2B in the presence of DNA damage, supporting the potential crosstalk between H2A and H2B ubiquitination [25]. Overall, loss of RNF20-RNF40 gives rise to increased damage and renders cells sensitive to DSB-inducing agents [90]. These findings mimic those in yeast depleted for Bre1, the homolog to RNF20-RNF40 [93,94,95]. Similarly, use of H2BK120/125R, an H2B mutant that cannot be ubiquitinated (K125 can be ubiquitinated in the event that K120 is unavailable) also delayed resolution of DSB foci. These cells, too, have reduced HR and NHEJ capacities and were more sensitive to IR and mitomycin C. The combination of RNF20 depletion and mutant H2B does not further exaggerate these defects, suggesting that RNF20 does in fact mediate its effects on DSB repair through H2B ubiquitination [90,91]. Taken together, these data underscore the importance of H2B ubiquitination in the repair of DSBs.

It was initially speculated that loss of H2BK120ub1 may abrogate DSB repair by impairing transcription of genes involved in these repair processes; however, expression of several key repair factors is not influenced by loss of RNF20-RNF40 [90,91,96]. Additionally, inhibiting transcription has no impact on RAD51 or BRCA1 foci [90]. Interestingly, there are several current hypotheses that propose a non-canonical role for H2BK120ub1 in DSB repair. H2BK120ub1 likely facilitates the decondensation of chromatin to allow key repair factors access to sites of damage [91,97]. This idea is seemingly counterintuitive when considering the simultaneous presence of H2AK119ub1, a repressive mark, at sites of DSBs. However, while H2BK120ub1 is deposited in close proximity to sites of DSBs to facilitate chromatin accessibility, H2AK119ub1 decorates areas more distal to DSBs in order to repress transcription [91,92]. Furthermore, promoting chromatin decondensation by treatment with chloroquine, trichostatin A, or hypotonic buffer restores IRIF formation in RNF20-depleted cells [90]. It has been proposed in yeast that monoubiquitination of H2B facilitates the accumulation of downstream repair factors at sites of breaks through altering histone dynamics and promoting nucleosome disassembly, reminiscent of its known role in transcriptional elongation [94,98,99]. In support of this, H2B ubiquitination has been shown to influence crosstalk with other histone marks including H4K20 methylation and H4K9 methylation that in turn may impact the downstream recruitment of DSB factors [94,95].

### 2.5. The Potential Contribution of H3 and H4 Ubiquitination to DSB Repair

In contrast to H2A and H2B, H3 and H4 ubiquitination are largely understudied in terms of DSB repair. Both H3 and H4 can be ubiquitinated in vitro and in vivo, albeit at relatively low levels (0.3% of total H3 and 0.1% of total H4) [100,101]. The CUL4-DDB1-ROC1 complex, which is conserved from yeast to humans, was identified as a bona fide histone E3 ligase with preferential activity towards H3 and H4 in vivo [101,102]. While Wang and colleagues studied the role of this complex in response to UV radiation [101], work done in fission yeast provides some insight into the potential for this complex to function in DSB repair as well. The yeast homolog to DDB1, *ddb1* was identified in a screen for regulators of genomic stability following DSB induction. *ddb1* deletion mutants had prolonged unrepaired damage and heightened sensitivities to IR and bleomycin. Ultimately, Ddb1 was proposed to promote efficient HR repair in order to maintain genomic stability [103]. While here a non-histone target of the CUL4-DDB1-ROC1 complex was responsible for this phenotype, this poses the idea that this complex may be able to regulate different types of ubiquitination modifications (i.e., monoubiquitination and polyubiquitination) on both histone and non-histone substrates [103]. While the impact of the CUL4-DDB1-ROC1 complex on histone ubiquitination has not directly been tested in the context of DSBs, this evidence makes such a link plausible. 

More convincing data regarding the role of H4 ubiquitination and the DSB response have been established. The E3 ligase BBAP directly interacts with and monoubiquitinates H4 in vivo, specifically at H4K91 (H4K91ub1) [104]. Expression of WT BBAP protected cells from hydroxyurea or doxorubicin treatment compared to those expressing a catalytically inactive mutant, implicating a requirement for BBAP E3 ligase activity. Additionally, an H4K91 mutant yeast strain had increased sensitivity to hydroxyurea and doxorubicin [105], pointing to a role for H4K91ub1 in regulating the response to DSB-inducing agents. Interestingly, BBAP could not be detected at sites of IRIF, implying a transient interaction. Furthermore, while the accumulation of early DSB responders including γH2AX and MDC1 were not impacted by depletion of BBAP, 53BP1 kinetics were significantly delayed and a minor decrease in BRCA1 foci was noted. In an effort to investigate the mechanism underlying defective 53BP1 clearance, levels of H4 monoubiquitination as well as H4K20 methylation, a mark that influences 53BP1 binding to sites of DSBs, were monitored. While treatment with doxorubicin increased BBAP expression and monoubiquitinated H4, H4K20 mono and dimethylation increased as well. Interestingly, upon knockdown of BBAP, levels of these methyl marks decreased and were accompanied by a corresponding increase in H4K91 acetylation. The current working model suggests that ubiquitination at H4K91 in response to DSBs may license chromatin for subsequent PTMs including H4K20 methylation [104]. 

## 3. The Reversal of Histone Ubiquitination at DSBs 

Numerous histone-modifying DUBs have been implicated in the repair of DSBs (Figure 1, Table 1). While a bulk of the literature has focused on reversal of H2A ubiquitination, there is a growing interest in how removal of H2B ubiquitination impacts the repair of DSBs. To date, no deubiquitinating enzymes have reported activity towards H3 or H4, warranting future investigations to uncover how these modifications are influenced by DSBs and how their removal impacts repair. As the DUBs introduced below have several possible substrates and functions in DSB repair, they have been listed in alphabetical order. 

### 3.1. BAP1: Polycomb Activity at DSBs

In drosophila, the polycomb repressive deubiquitinase complex (PR-DUB) consists of the deubiquitinating enzyme Calypso and adaptor protein ASX. Together, this complex controls polycomb-mediated transcriptional repression by catalyzing the removal of H2AK118 monoubiquitination (H2AK119ub1 in mammals). PR-DUB is evolutionarily conserved in mammals and is comprised of BAP1 and one of three mammalian ASX homologs, ASXL1/2/3. BAP1 is a member of the UCH family of DUBs and is localized to the nucleus [161]. Similar to PR-DUB, BAP1 exhibits activity towards monoubiquitinated H2A with a strong preference for K119 as opposed to K13/15 [111,112]. Global changes in H2AK119ub1 are dependent on both BAP1 catalytic activity and the presence of an ASXL adaptor protein [71,111,162]. As a phosphorylation target of ATM, BAP1 has been linked to DSB repair [71,106,107,108]. Chromatin bound BAP1 is enriched following the induction of DSBs and colocalizes with γH2AX [71,107,109]. Importantly, ASXL1 can also be detected at sites of damage, indicating that BAP1 may be catalytically active in this context [71]. Chromatin immunoprecipitation experiments have revealed a negative correlation between BAP1 binding and levels of H2A/H2AX monoubiquitination at sites flanking DSBs [71,107]. siRNA-mediated knockdown of BAP1 causes DSB repair defects as evidenced by decreased IR-induced BRCA1, RAD51, and RPA foci [71,107,109]. Notably, 53BP1 foci are not altered in the BAP1-deficient setting, further owing to the specificity of BAP1 for H2AK119ub1 and supporting a role for BAP1 in HR regulation [107,110]. In line with this, BAP1 knockout cells are more sensitive to IR and PARP inhibitors, both characteristic of an HR deficiency [71,107]. Use of GFP reporter assays has confirmed decreased HR efficiency upon knockdown of BAP1 but no impact on NHEJ [71]. While direct evidence connecting BAP1 activity at H2AK119ub1 to the observed HR defects is lacking, it is plausible that BAP1 may regulate transcription in the vicinity of DSBs or directly impact the transcription of repair effectors themselves. BAP1 may also mediate regulation of chromatin accessibility through H2AK119 deubiquitination, to in turn impact recruitment of repair factors [107]. These hypotheses are appealing considering other polycomb-group proteins and chromatin modifiers have been observed at sites of DSBs [64,68,163,164]

### 3.2. BRCC36: Restricting End Resection through Removal of K63-Linked Polyubiquitin Chains

BRCC36 is a member of the JAMM family of DUBs that has been detected at IRIF [109]. While BRCC36 was originally deemed catalytically inactive [119], it is now appreciated that BRCC36 requires association with scaffolding proteins and the formation of multi-subunit complexes for DUB activity [113]. Abraxas (CCDC98), a nuclear protein, anchors BRCC36 to the BRCA1-A complex along with BRCA1, MERIT40, BRCC45, and RAP80 [17,35,113,165]. RAP80 binds to K63-linked polyubiquitin chains at H2A-type histones and is responsible for BRCC36 localization to sites of damage [35,120]. Together this complex restricts end resection, and thus HR, during the response to DSBs [40]. Increased sensitivity to IR upon loss of BRCC36 can be restored upon co-depletion of RNF8 [114,115,119], suggesting BRCC36 counteracts RNF8/RNF168. Indeed, the role of BRCC36 is to catalyze removal of K63-linked polyubiquitin chains from H2A-type histones. K63-linked polyubiquitin foci are elevated both at baseline and post-IR in BRCC36 depleted cells and can be restored upon rescue with catalytically active BRCC36 [65,113,115,116]. Interestingly, while cells depleted for BRCC36 or those that express catalytically inactive BRCC36 have increased 53BP1 foci [115], an increase in HR activity is observed [40,116]. The current understanding of this seemingly paradoxical phenomenon suggests that depletion of BRCC36 leads to disruption of BRCA1-A complex formation at sites of DSBs [116]. BRCC36 depletion leads to an increase in phosphorylated RPA (a readout for end resection), supporting the role of BRCA1-A in preventing hyperactivation of this repair pathway. The underlying mechanism supporting this model remains unclear. It is possible that BRCC36 regulates RAP80, Abraxas, and MERIT40 expression levels or the accumulation of K63-linked ubiquitin conjugates in the absence of BRCC36 is inhibitory to RAP80 recruitment [116].

BRCC36 does not solely associate with the BRCA1-A complex. In the cytoplasm, BRCC36 forms a distinct complex known as BRISC (BRCC36 isopeptidase complex) alongside KIAA0157 (Abro), BRCC45, and MERIT40 [117,118]. While this complex can deubiquitinate K63-linked polyubiquitin chains in the cytoplasm, it does not participate in the repair of DSBs in the nucleus. Interestingly, depletion of KIAA0517 leads to increased BRCA1-A complex association in the nucleus [113], which may have important implications in the DSB response. For future studies regarding BRCC36, it is important to consider these complexes as they do not only regulate BRCC36 cellular localization but activity as well.

### 3.3. DUB3/USP17L2: Counteracting RNF8/RNF168-Mediated IRIF

An expression vector screen aiming to pinpoint novel regulators of the DSB response identified DUB3 [121]. DUB3, also known as USP17-like 2 (USP17L2), is a member of the USP family of DUBs. While colocalization of DUB3 with known markers of DSBs has not been directly assessed, DUB3 is predominantly detected in the nucleus and bound to chromatin. Overexpression of DUB3 does not impact the presence of γH2AX or MDC1 at IRIF, yet the kinetics of γH2AX clearance are impaired and cells are more sensitive to camptothecin treatment [121]. This not only suggests a role for DUB3 in promoting effective and timely DSB repair, but that its activity is downstream of MDC1. As expected, overexpression of RNF8 and RNF168 led to increased levels of ubiquitinated H2A. However, when DUB3 was co-expressed with these E3 ligases, levels of H2Aub were reversed, implying that DUB3 may antagonize this step of the DSB repair pathway. In line with this, DUB3 was found to interact with purified H2AX, a target of RNF8/RNF168, and H2AX ubiquitination levels were dependent on DUB3 catalytic activity [121]. Overexpression of DUB3 decreased monoubiquitination of γH2AX in the absence of an exogenous DNA damage source as well as post-IR [33]. Interestingly, RNF168 recruitment to IRIF, but not RNF8, was impaired upon DUB3 overexpression. This initially implied that DUB3 may counter RNF8 activity at H2A/H2AX, however it is plausible that DUB3 simply limits the ability for RNF168 to recognize these ubiquitin conjugates [34]. Further supporting a role for DUB3 in antagonizing RNF8/RNF168, overexpression of DUB3 led to a reduction in 53BP1 and BRCA1 foci [121]. Despite the above evidence, an independent group identified DUB3 in an RNAi screen where knockdown led to decreased BRCA1 and RAD51 foci, meaning DUB3 may promote HR as well [107]. Further studies are needed to solidify the exact role of DUB3 in the DSB response. 

### 3.4. MYSM1/2A-DUB: An Underexplored H2AK119ub1 Modifier

MYSM1 is a member of the JAMM family of DUBs. The presence of a SANT domain and a SWIRM domain, both of which allow for interactions with histones and/or DNA, highlights the possibility for MYSM1 to modify histone substrates [122,123,124]. In fact, MYSM1 is localized primarily to the nucleus and can directly interact with both H2A and H2B [124]. Although H2B ubiquitination is seemingly unaffected by overexpression of MYSM1, both in vitro and endogenous H2A ubiquitination was decreased. Regulation of H2A ubiquitination, specifically mapped to K119, is dependent on MYSM1 catalytic activity [124]. While it is no surprise that MYSM1 activity towards H2AK119ub1 has been linked to transcriptional regulation [124,125,166], this DUB has also been implicated in the DSB response. MYSM1 localizes to sites of DSBs, and when lost, there is sustained DNA damage [109]. In mice, depletion of MYSM1 increases sensitivity to radiation [126]. Similar to BAP1, MYSM1 does not impair 53BP1 foci following irradiation, suggesting that this DUB may favor HR [110]. The underlying mechanism linking MYSM1 to the DSB response has not been fully elucidated and the possibilities described above for BAP1 regarding transcriptional regulation and chromatin accessibility hold true here as well.

### 3.5. POH1/PSMD14: A Non-Degradative Function of the Proteasome

The JAMM family member POH1/PSMD14 was identified as a novel regulator of DSB repair, where loss led to enlarged 53BP1 foci following the induction of damage [109,129]. This phenomenon is dependent on POH1 DUB activity, as overexpression of WT but not catalytically inactive POH1 can reduce the size of these foci [129]. POH1 is an integral subunit of the 26S proteasome, specifically the 19S lid [127,128]. POH1 canonically facilitates cleavage of K48-linked polyubiquitin chains to couple deubiquitination to degradation [128]. With this in mind, it was initially suspected that POH1 may regulate 53BP1 degradation. Surprisingly, however, modulating POH1 had no effect on 53BP1 half-life or expression levels [129]. A potential explanation for this was offered when POH1 was purified as a K63-linkage specific DUB [117]. Accordingly, POH1 depletion led to persistent ubiquitin-conjugates at sites of DSBs, specifically K63-linked ubiquitin. POH1 is localized to sites of DSBs and can directly antagonize RNF8/RNF168, which collaborate to deposit K63-linked polyubiquitin chains at H2A-type histones in response to damage [129]. As mentioned above, K63-polyubiquitinated H2A is an essential recruitment platform for downstream mediators of the DSB response including 53BP1. Interestingly, knockdown of core subunits of the proteasome or use of the proteasome inhibitor MG132 also increased 53BP1 accumulation, suggesting that the entire proteasome may somehow participate in this non-canonical function. This echoes earlier work done in yeast that observed recruitment of the 26S proteasome to sites surrounding DSBs [130]. 

Interestingly, when co-depleted with BRCC36, another JAMM family member that displays K63-specificity, 53BP1 impairment was not exaggerated, suggesting these DUBs work in the same pathway. In fact, POH1 depletion alone has a greater impact on 53BP1 foci, suggesting POH1 may play additional roles in DSB repair [129]. To this end, POH1 has been suspected to regulate 53BP1 accumulation by removing RAP80 from sites of DSBs [131] or through regulation of JMJD2A [129]. Similar to 53BP1, JMJD2A contains a TUDOR domain that allows for recognition of H4K20 methyl marks. Similar to L3MBTL1 described above, JMJD2A is polyubiquitinated via RNF8 (K48-linkages) upon DNA damage and sent for proteasomal degradation, exposing H4K20 methylation to allow for 53BP1 recruitment. Although the underlying mechanism is not yet fully understood, POH1 catalytic activity was found to promote JMJD2A residence on chromatin [129]. These mechanisms are not necessarily mutually exclusive, and both point to a role for POH1 in restricting 53BP1. Interestingly, depletion of POH1 leads to a decrease in NHEJ despite increases in 53BP1 accumulation [129]. Co-depletion of POH1 and 53BP1 restored NHEJ, suggesting excessive 53BP1 can actually impair NHEJ and that POH1 regulates the efficiency of this process. Alternatively, POH1 has been implicated in the promotion of HR. While one group identified decreased BRCA1 foci upon depletion of POH1 [107], others have attributed POH1′s role in HR to relocalization of RAP80 in order to facilitate RPA and RAD51 loading on chromatin [129,131]. 

### 3.6. USP3: Antagonizing RNF8/RNF168 at H2A-Type Histones

USP3 is a chromatin bound member of the USP family that was isolated and characterized over 20 years ago [133]. USP3 has been implicated in protecting the genome from spontaneous damage both in vitro and in vivo. Knockdown of USP3 alone activates the DSB response, as determined by a number of factors including a delay in S-phase progression, an increased number of DNA breaks and structural abnormalities, ATM/ATR activation, and γH2AX staining [109,132,134]. Although USP3 localization to DSBs is most likely transient, ubiquitinated H2A and H2AX persist following IR upon USP3 depletion, suggesting USP3 may modify these substrates [134]. USP3 does in fact interact with ubiquitinated H2A-type histones, and mouse embryonic fibroblasts isolated from USP3-null mice express increased levels of ubiquitinated H2A [132]. In vitro, USP3 was specifically shown to deubiquitinate residues K13/15 on H2A and both K13/15 and K119 on H2AX [134,135]. Consistent with this observation, overexpression of USP3 does not impact RNF8 retention at sites of DSBs, however it does reduce RNF168 accumulation [18]. 53BP1 and RAP80 foci are also negatively regulated by USP3 catalytic activity [18,110,132,135,136]. Further solidifying a role for USP3 in the DSB response, USP3-null mice are more sensitive to total body irradiation when compared to their WT counterparts [132]. Interestingly, USP3 was a hit in an RNAi screen identifying DUBs that regulate H2BK120ub and 53BP1 foci [137]. While the activity of USP3 at this residue has not been thoroughly investigated in the context of the DSB response, USP3 has been shown to regulate levels of ubiquitinated H2B in vitro and in vivo (albeit to a lesser extent than ubiquitinated H2A) [110,132,134,137]. 

### 3.7. USP11: Promoting HR through Deubiquitination of γH2AX 

Several lines of evidence suggest USP11 activity facilitates functional HR. Not only was USP11 identified in a screen for PARP inhibitor sensitivity, but depletion of USP11 alters RAD51 foci formation and enhances sensitivity to other DSB-inducing agents as well [33,109,138]. Use of a GFP reporter system confirmed that HR efficiency is compromised upon USP11 depletion. Mechanistically, USP11 interacts with and deubiquitinates γH2AX [33]. This same group identified ubiquitinated γH2AX as substrates for RNF8/RNF168, which adds USP11 to the growing list of DUBs that may counter the activity of these E3 ligases [33]. In support of this, RNF8 depleted cells showed a delay in IR-induced 53BP1 foci, as expected, while depletion of USP11 led to hyperaccumulation of ubiquitin conjugates and 53BP1 [33]. Although this work argues that USP11 is specific for ubiquitinated γH2AX, others have shown USP11 activity towards H2AK119ub1 as well as H2BK120ub1 [137]. In these studies, the defects in DSB repair and IR sensitivity observed upon loss of USP11 were restored when USP11 was co-depleted with either BMI1 or RNF20 (involved in catalyzing H2AK119ub1 and H2BK120ub1, respectively) [137]. This reveals several additional mechanisms by which USP11 may influence DSB repair including transcriptional regulation and chromatin conformation. As such, USP11 has been shown to interact with known transcriptional regulators including the nucleosome remodeling deacetylase (NuRD) complex. The interactions between USP11, NuRD, and chromatin are increased following IR, suggesting USP11 may be recruited to sites of DSBs to aid in the regulation of gene expression. Additionally, USP11 depletion led to increased sensitivity to micrococcal nuclease following IR when compared to controls, and therefore may be able to modulate DNA repair factor access to chromatin as well [137]. From this body of work, it is clear that the role of USP11 in the repair of DSBs is much more complex than what is currently understood.

### 3.8. USP16/Ubp-M: Coordinating Transcriptional Repression at Sites Neighboring DSBs

USP16/Ubp-M is an H2A-modifying member of the USP DUB family [139]. HERC2, which was previously shown to associate with both RNF8 and RNF168 to promote K63-linked ubiquitination, was identified as an interacting partner of USP16 [140,141,142]. Irradiation-induced damage increases expression of USP16 in a HERC2-dependent manner. Loss of HERC2 decreases both USP16 and levels of ubiquitinated H2A, which can be partially restored by rescuing USP16 [142]. Interestingly, USP16 overexpression does not impair RNF8 or RNF168 localization [136,142], suggesting two possibilities: USP16 activity is independent of these ligases or downstream of them. As mentioned above, transcription is locally repressed in the presence of DSBs [65]. This is dependent on ATM and correlates with H2AK119ub1 levels. Accordingly, inhibition of ATM reduces levels of H2AK119ub1 to restore transcription. In this same study, H2AK119ub1 levels remained elevated and transcription was restricted upon knockdown of USP16, even in the presence of an ATM inhibitor. Expressing an siRNA-resistant USP16 restored transcription levels in this system, implicating USP16 in H2AK119ub1-mediated transcriptional regulation at DSB sites. In vitro deubiquitination assays using H2A mutants (K13/15R and K118/119R) reveal that USP16 can act towards both residues [142]. Although this idea has been challenged [136], USP16 activity towards H2AK13/15 would open up new possible roles for this DUB in DSB repair. Currently, controversial evidence exists regarding the impact of USP16 on 53BP1 foci [110,136]. Although confirmation of the relevant substrate(s) of USP16 for DSB repair will help clarify these functional consequences, the most recent study published on USP16 challenges the direct role of this DUB in the response to DSBs [143]. Here, USP16 was found in the cytoplasm throughout the cell cycle and surprisingly did not relocate to the nucleus upon induction of DNA damage. This group did, however, observe catalytically inactive USP16 retained in the nucleus. Remarkably, depletion of PRC1 subunits abolished this retention, and the authors speculate USP16 in this scenario is trapped to substrates like H2AK119ub1. This is a thought-provoking piece of data that warrants further investigation of USP16 activity at H2AK119ub1 and how it relates to the response to DSBs [143].

### 3.9. USP22: Context-dependent Regulation of NHEJ by H2BK120ub1

USP22, in association with adaptor proteins ATXN7, ATXN7L3 and ENY2, comprises the DUB module of the SAGA (SPT-ADA-GCN5 acetyltransferase) chromatin modifying complex. While this module is well established as a critical regulator of H2BK120ub1 to facilitate transcriptional elongation in mammals, an in vitro screen identified SAGA components as regulators of CSR, which again involves the generation of programmed DSBs that are primarily repaired via NHEJ [144]. Depletion of ATXN7, ENY2, or USP22 led to decreased levels of CSR in vitro, which was later confirmed in a B-cell specific USP22 knockout mouse [145]. This suggests a role for USP22 in regulating NHEJ. Interestingly, USP22 does not impact 53BP1 following IR [110]. In an effort to better understand the contribution of USP22 to DSB repair, H2BK120ub1 levels were monitored following IR. Levels of this mark were increased and even more so in the absence of USP22 [144]. This is at odds with recent reports showing ablation of USP22 does not increase global levels of H2BK120ub1 due to redundancy with homologs USP27X and USP51 [146,147,148], although this discrepancy may be attributed to differences in cell types and context-dependent roles for these DUBs. USP22 depleted cells also exhibited impaired γH2AX kinetics [145], challenging previous reports that H2B ubiquitination does not influence H2AX PTMs [91]. Despite the evidence suggesting H2BK120ub1 regulates transcription at sites of DSBs, USP22 has not been directly studied in this context. Additionally, USP22 has been shown to deubiquitinate H2A, which represents another potential substrate for USP22 in DSB repair [145,146,149,150].

### 3.10. USP26, USP29, USP37: Functionally Redundant DUB Activity towards H2A

Although USP26 was originally thought to be a testis-specific homolog of USP29 [110,151,152], more recent evidence detecting USP26 in a variety of human cell lines and tissues has contradicted this classification [153,154,155]. USP26 and USP29 are now recognized as retrogenes of a third USP family member, USP37. Each of these DUBs have been implicated in the repair of DSBs. Overexpression of USP26, USP29, or USP37 does not impact γH2AX, MDC1, or RNF8 accumulation at DSB sites, suggesting they act downstream of these factors. In line with this, overexpression of these DUBs suppresses both RNF168 and 53BP1 foci formation [110,154]. USP26 and USP37 have additionally been shown to impair accumulation of RAP80 and BRCA1 [154]. Collectively, these data suggest that USP26, USP29, and USP37 can reverse RNF8/RNF168-mediated ubiquitination of H2A. USP29 has been shown to reduce levels of monoubiquitinated H2A [110], while USP26 and USP37 have more specifically been shown to reverse ubiquitination catalyzed by RNF8/RNF168 as opposed to K48-linked polyubiquitin chains or H2BK120ub1 [154].

Highlighting the functional redundancy of these DUBs, co-depletion of USP26 and USP37 does not further exaggerate DSB repair defects when compared to single depletions. Interestingly, in contrast with initial findings assessing overexpression of these DUBs, knockdown of either USP26 or USP37 impairs accumulation of key HR factors including RAD51 as well as PALB2, CtIP, and RPA [154]. Cells depleted for these DUBs are hypersensitive to PARP inhibition as well as IR, also indicative of an HR deficiency. Use of GFP reporter assays has confirmed reduced HR efficiency when these DUBs are depleted. This mimics the HR defects observed upon overexpression of RNF8/RNF168, emphasizing the importance of proper H2A ubiquitination levels for regulation of this repair pathway. USP26 and USP37 activity towards H2A is suspected to antagonize BRCA1-A complex formation at sites of DSBs in order to promote HR, suggesting specificity for K63-linked polyubiquitin chains. In support of this, depletion of either USP26 or USP37 led to increased ubiquitination and RAP80 accumulation, as well as larger and more extensive BRCA1 foci. Moreover, co-depletion of either USP26 or USP37 and RAP80 was able to restore levels of HR. While the specific role of USP29 in regulating HR has not been investigated, it is possible USP29 shares these functions given its similarities to USP26 and USP37. Unlike USP26 and USP37, however, USP29 displays activity towards H2B ubiquitination as well as non-histone substrates and may therefore have functions independent of its homologs [110].

### 3.11. USP44: Negative Regulation of Downstream Repair Machinery

USP44 has been well-established as a modifier of H2B ubiquitination [110,156,157]. Similar to many DUBs that require association with adaptor proteins for catalytic activity, recombinant USP44 is inactive in isolation. As a central component of the nuclear receptor corepressor (NCoR) complex, USP44 has been shown to coordinate deubiquitination of H2BK120ub1 and transcriptional repression [157]. Surprisingly, whether USP44 activity towards H2BK120ub1 impacts transcription in response to DSBs remains unanswered.

Live cell imaging has revealed rapid USP44 localization to laser-induced DSBs [109]. At later time points, however, USP44 can no longer be visualized [110], suggesting USP44 participates early on in the repair process. Further insight into the precise activity of USP44 at DSBs comes from studying its catalytically inactive form. In an RNF8/RNF168-dependent manner, catalytically inactive USP44 can be observed at IRIF >1 h post laser-induced damage. This suggests that RNF8/RNF168-mediated ubiquitination events (i.e., at H2AK13/15, are responsible for recruitment of USP44 to damaged sites) [110]. Furthermore, overexpression of USP44 impairs RNF168, 53BP1 and RAP80 foci, all of which can be directed to DSBs by ubiquitinated H2A. USP44 is therefore not only recruited to damage by ubiquitinated H2A but may directly catalyze the removal of these marks to impact downstream repair factors as well. Ectopically expressed USP44 reduces levels of monoubiquitinated H2A as well as higher-order chains [110], and has been speculated to deubiquitinate γH2AX [135], although the requirement for adaptor proteins has not been investigated in these contexts. Importantly, while USP44 overexpression is able to negatively regulate DSB repair factors, knockdown of this DUB only mildly impacts H2A ubiquitination levels and the response to DSBs [110]. This implies functional redundancy amongst USP44 and other H2A-targeting DUBs and warrants additional studies on the role of endogenous USP44.

### 3.12. USP48: Novel DUB Activity at H2AK125/127/129

Due to its ability to interact with ubiquitinated nucleosomes [167], USP48 was included in a screen for DUBs that specifically counteract BRCA1 E3 ligase activity at H2A [158]. In this screen, USP48 demonstrated a preference for nucleosomal H2A ubiquitinated by BRCA1-BARD1 at K125/127/129 over H2A ubiquitinated by RNF168 or PRC1. USP48 activity is most robust when additional, auxiliary ubiquitin molecules are present on the BRCA1 site or K119. Interestingly, although BRCA1-BARD1 is suspected to catalyze K6-linked polyubiquitin chains, USP48 only showed activity towards K27-linked di-ubiquitin in vitro [158].

Given the role of BRCA1-BARD1-mediated ubiquitination of H2A in the repair of DSBs, this group set out to determine the role of USP48 in this context. Following IR, USP48 localizes to sites of damage along with 53BP1 and BRCA1. Importantly, this localization was reduced when BRCA1 was depleted. Consistent with a role for USP48 in antagonizing BRCA1-BARD1, USP48 depleted cells have higher numbers of RAD51 and RPA foci and these foci were more intense following IR. Rescue with WT USP48 restored foci levels while a catalytically inactive USP48 was unable to do so. USP48 depleted cells also had longer resection lengths compared to control cells, which again were restored upon expression of WT USP48. As mentioned earlier, SMARCAD1 is critical for BRCA1-BARD1-dependent regulation of DNA end resection. As expected, the same group found that depletion of BRCA1 or SMARCAD1 led to shortened resection lengths which were unaffected by co-depletion with USP48. Once again in line with a role for USP48 in regulating end resection, there was a greater spread of 53BP1 in USP48 depleted cells. An overall model has been proposed in which USP48 limits BRCA1-BARD1 ubiquitination of H2AK125/127/129. This in turn limits the recruitment and activity of SMARCAD1 and prevents repositioning of 53BP1, ultimately restricting end resection [158].

Of note, the panel of DUBs in this initial screen was limited to select H2A-modifying DUBs (USP3, USP16, BAP1), DUBs involved in the DNA damage response (USP1, USP11, USP7, USP15, USP12) and USP48. Therefore, it is likely that additional DUBs not included in this study also have activity towards this mark and may be implicated in the repair of DSBs.

### 3.13. USP51: Fine-Tuning the Repair of DSBs

Increased spontaneous DNA damage foci as evidenced by ubiquitin conjugates, BRCA1, and 53BP1 staining were observed in cells depleted for USP family member USP51. Foci were restored upon rescue with shRNA-resistant USP51, but this was dependent on USP51 catalytic activity. USP51 most likely works downstream of RNF168 in the DSB repair pathway, considering its overexpression does not impair the accumulation of γH2AX, MDC1, or RNF168 itself. Conversely, USP51 overexpression does lead to reduced RNF169, 53BP1, and BRCA1 foci, all of which are recruited to RNF168 targets. Based on these observations, Wang and colleagues were led to investigate USP51 as an H2AK13/15 specific DUB [136]. In support of this idea, purified USP51 was shown to interact with H2A-H2B dimers and in vitro deubiquitination assays revealed that catalytically active USP51 decreases levels of H2AK15ub1. Notably, USP51 also displayed activity towards K27-linked polyubiquitin chains [136]. This non-canonical mark is also deposited by RNF168 and points to an interesting, but currently unexplored, avenue by which USP51 may impact the repair of DSBs. To understand the role of USP51 in a more relevant setting, cells were irradiated and fractionation experiments were performed. The amount of USP51 in the chromatin fraction was decreased following IR treatment, but began to increase 1 hr post-treatment. This negatively correlated with levels of γH2AX and H2AK15ub1, which both rapidly increased following IR but began to decrease at the 1 hr time point. When USP51 was depleted, H2AK15ub1 foci accumulated much more rapidly and persisted in irradiated cells, which was accompanied by lasting RNF168 and 53BP1 foci. Although depletion of USP51 led to increased repair by HR and NHEJ, these cells had increased levels of damage and were more sensitive to IR. While seemingly counterintuitive, hyperactivation of HR can contribute to extended end resection and therefore cause unwarranted damage. Likewise, excessive RNF168 or 53BP1 has been shown to promote mutagenic NHEJ [159]. Interestingly, USP51 overexpression also increases sensitivity to IR, suggesting DSB repair needs to be maintained in a delicate balance. All in all, the following model has been proposed to explain how USP51 can fine-tune these repair mechanisms: upon induction of DSBs, USP51 is released from chromatin, allowing for monoubiquitination of H2AK13/15 and subsequent recruitment of downstream repair factors such as BRCA1 and 53BP1. Following the repair of damage, USP51 is redirected back to chromatin (by a yet to be determined mechanism) to deubiquitinate H2AK15ub1 and ultimately terminate this response. 

It is likely that USP51 has additional substrates that contribute to the DSB response. In addition to the K27-linked polyubiquitin chains mentioned above, our lab and others have identified USP51 activity towards H2AK119ub1 and H2BK120ub1 [146,160]. While USP51 activity towards H2AK119ub1 went undetected by Wang et al., this study did not address the role of adaptor proteins ATXN7L3 and ENY2 that are required for USP51 catalytic activity [146]. Therefore, it is possible that USP51 influences transcription at DSBs to in a manner similar to other H2AK119ub1 or H2BK120ub1-modifying DUBs. 

## 4. Beyond Direct Histone Ubiquitination and Deubiquitination

While in this review we have primarily focused on direct histone ubiquitination and deubiquitination events, DUBs can also indirectly influence histone PTMs in response to DSBs. For example, USP7 is recruited to sites of DSBs, where it stabilizes RNF168 and PRC1, thereby influencing H2A ubiquitination levels [168]. USP7 additionally regulates the DSB response by stabilizing PHF8, a histone demethylase, in order for proper recruitment of DNA repair factors BLM and KU70 to damage sites [169]. USP7 has also been shown to more directly regulate levels of H2A and H2B ubiquitination and may therefore have additional roles in DSB repair that have yet to be uncovered [170,171]. DUBs can also modify non-histone substrates to influence the DSB response. USP4, for example, interacts directly with end resection factors CtIP and MRN [172], USP21 deubiquitinates and stabilizes BRCA2 to promote RAD51 binding and successful HR [173], and USP10 deubiquitinates and stabilizes p53 to promote its nuclear localization and apoptosis in response to DSBs [174]. DUBs may also contribute to the DSB response through non-catalytic mechanisms. For example, OTU family member OTUB1 binds to E2 conjugating enzymes to prevent their activity [175]. 

It is also important to note that ubiquitination cooperates with other histone PTMs including sumoylation, phosphorylation, methylation, acetylation, and PARylation in order to elicit an efficient response to damage. In this review we discussed several relevant examples of this crosstalk, including the ability for both ubiquitination and sumoylation to recruit RAP80 and therefore the BRCA1-A complex to DSBs. Loss of sumoylation at sites of damage has also been shown to impair K63-linked polyubiquitin chains as well as RNF168 and 53BP1 recruitment [36,37]. While ATM-mediated phosphorylation of MDC1 promotes the recruitment of the E3 ubiquitin ligase RNF8, RNF8 in turn can positively feedback to promote ATM activation and subsequent phosphorylation events. This is thought to be through regulating histone acetylation and chromatin conformation [176]. Interestingly, activity of histone acetyltransferase TIP60 and chromatin remodeler p400 has been shown to reciprocally promote RNF8-mediated ubiquitination and therefore the recruitment of downstream repair factors [177], showcasing the interconnectedness of these modifications. PARylation has additionally been found to recruit the NuRD complex to chromatin to facilitate deacetylation. Loss of the NuRD complex in turn influences histone ubiquitination events at sites of DSBs [178]. Methylation and ubiquitination also act in concert to regulate DSB repair, the most notable example being the dual recognition of H2AK15ub1 and H4K20me2 by 53BP1 [41]. BBAP-mediated monoubiquitination of H4K91 may also prime histones for additional PTMs including H4K20 methylation [104]. A more comprehensive overview of these modifications is presented elsewhere (see [179,180]).

## 5. DUBs as Therapeutic Targets

The deregulation of DUBs has been implicated in several diseases including developmental disorders [181], neurodegeneration [182], autoimmune diseases [183], infectious diseases [184], and cancer. Impaired DSB repair contributes to genomic instability, a hallmark of cancer, and it is not a surprise that several DUBs involved in DSB repair have been implicated in carcinogenesis. Germline mutations in BAP1, a bona fide tumor suppressor, are associated with BAP1 tumor predisposition syndrome that puts carriers at a higher risk for several cancer types including uveal melanoma, malignant mesothelioma, clear cell renal cell carcinoma, and basal cell carcinoma [185]. Several other DUBs discussed in this review including DUB3/USP17L2, USP3, and POH1/PSMD14 have been implicated in cancer for their abilities to regulate migration, invasion, and drug resistance [186,187,188]. Further, USP22 has been identified as a component of an 11-gene “death from cancer” signature that predicts poor patient prognosis [189]. It is important to note that these specific examples have yet to be directly link to deubiquitination of histones in response to DNA damage. 

DUBs are druggable enzymes and are increasingly being recognized as potential therapeutic targets. POH1 depletion, for example, has been shown to impair multiple myeloma cell proliferation [190]. Likewise, knockdown of BRCC36 increases sensitivity of breast cancer cells to IR [114] and knockdown of USP11 sensitizes cells to PARP inhibition [138]. Collectively, these data suggest that inhibiting DUBs implicated in DSB repair is a viable therapeutic strategy to sensitize cells to DNA damaging agents. This has prompted the development of a myriad of small molecule DUB inhibitors [191]. Although inhibiting DUB activity may offer more specificity and therefore lower toxicity in comparison to traditional proteasome inhibitors, a major challenge for the development of DUB inhibitors comes from the structural similarities between the catalytic domains of DUB family members. Furthermore, as many DUBs require interaction with different adaptor proteins for activity, molecules targeting these interactions may offer a better strategy for blocking specific DUB activity without interfering with a common catalytic domain. Nevertheless, specific inhibitors have been developed for USP7 and successfully used in preclinical models [192,193]. The success with USP7 inhibitors is encouraging and warrants future efforts for developing inhibitors for histone-modifying DUBs involved in DSB repair that can be used as anticancer therapies.

## 6. Concluding Remark

The timely and efficient repair of DSBs requires the close coordination of several cellular processes including the cell cycle, transcription, and the accumulation of repair machinery. These processes are governed by complex signaling cascades involving a series of well-documented phosphorylation and ubiquitination events that are critical for maintaining genomic stability. As this review has outlined, ubiquitination events are fundamental in the response to DSBs not only as recruitment platforms for downstream DSB effectors (H2AK13/15), but as critical modulators of gene expression (H2AK119), determinants of repair pathway choice (H2AK125/127/129), and regulators of chromatin architecture (H2BK120). 

The reversal of histone ubiquitination by DUBs is equally as important, and there is a growing body of evidence detailing how these enzymes contribute to an effective response to damage. There are several important considerations regarding DUBs, the first being that many do not act in isolation. The requirement for adaptor proteins and complex formation is not unique to DUBs involved in DSB repair and importantly, these associations can influence localization and enzymatic activity. Second, a majority of histone-modifying DUBs act on the same substrates, namely H2AK13/15, H2AK119, H2AK125/127/129, and H2BK120. This suggests either functional redundancy or context-dependent activities. The former is illustrated by the roles of USP26, USP29, and USP37, while the ability of USP22 to specifically alter global H2BK120ub levels in B cells supports the latter. As mentioned above, there are indirect mechanisms by which DUBs may influence histone ubiquitination and the crosstalk with additional PTMs. Ultimately, the role of these enzymes in DSB repair may be exploited therapeutically in several disease settings including cancer. The list of ubiquitination machinery provided in this review is most likely far from exhaustive, especially considering no H3 or H4 DUBs have been identified, but nonetheless highlights the importance of these modifications in the response to DSBs.

## Figures and Tables

**Figure 1 cells-09-01699-f001:**
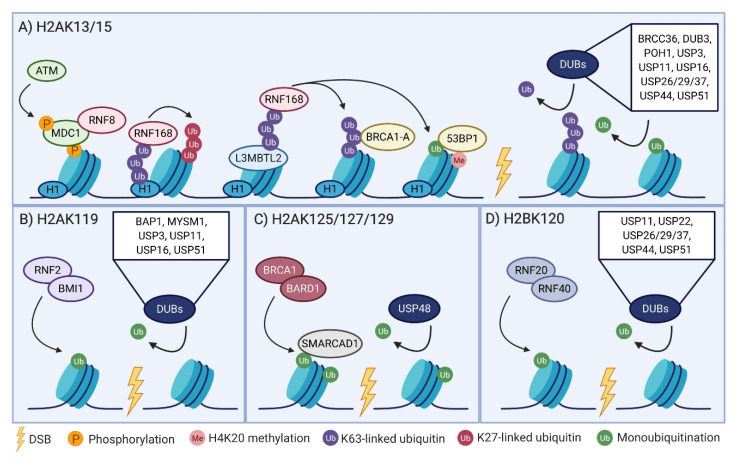
Mediators of ubiquitination and deubiquitination following a double strand break (DSB). (**A**) DSB induction, represented by a lightning bolt, leads to phosphorylation of H2AX (γH2AX) by ataxia-telangiectasia mutated (ATM) kinase. γH2AX recruits MDC1, which in turn recruits RNF8. RNF8-mediated ubiquitination of H1 or L3MBTL2 has been proposed to recruit RNF168. The concerted actions of RNF8 and RNF168 result in the mono (green), K63-linked (purple), and K27-linked (dark pink) ubiquitination of lysines 13 and 15 on H2A-type histones. H2AK13/15 ubiquitination acts as a recruitment platform for downstream mediators of the DSB response including the BRCA1-A complex and 53BP1, the latter dually recognizing ubiquitin and H4K20 methylation (light pink). Deubiquitinating enzyme (DUB) activity towards H2AK13/15 influences the recruitment of these factors and ultimately impacts the efficiency of DSB repair. (**B**) A PRC1 catalytic heterodimer, most notably RNF2 and BMI1, catalyzes H2AK119 monoubiquitination. This mark facilitates transcriptional silencing and its removal by various DUBs can influence gene expression at sites surrounding DSBs. (**C**) Ubiquitination at H2AK125/127/129 is catalyzed by BRCA1-BARD1 in response to DSBs. This mark is subsequently recognized by SMARCAD1, which facilitates the repositioning of 53BP1 to promote DNA end resection and thus homologous recombination (HR). USP48 is currently the only identified DUB with specificity towards this residue. USP48 activity is most robust in the presence of auxiliary ubiquitin either at K119 or the BRCA1 site. While monoubiquitination is depicted in this figure, polyubiquitin chains may be synthesized at the BRCA1 site as well. (**D**) Monoubiquitination at H2BK120 is deposited by RNF20-RNF40. The reversal of this mark by several DUBs has been reported and may influence the accessibility of chromatin for downstream DSB mediators. While ubiquitination of H3 and H4 occurs, future studies warrant the investigation of these marks and the DUBs that reverse them in the context of DSB repair. Figure created with BioRender.com.

**Table 1 cells-09-01699-t001:** Histone-modifying DUBs, the complexes they form, and their putative roles in the repair of DSBs.

DUB	DUB Family	Associated Adaptor Proteins/ Complexes	Histone Substrates	Functions in DSB Repair	Refs.
BAP1	UCH	ASXL1/2/3 (PR-DUB complex)	H2A(X)K119ub1	Promotes HR, transcriptional regulation ^#^, chromatin accessibility ^#^	[71,106,107,108,109,110,111,112]
BRCC36	JAMM	Abraxas, RAP80, BRCA1 (BRCA1-A complex) KIAA0517, BRCC45, MERIT40 (BRISC complex)	H2A(X) K63-linked chains	Counteracts RNF8/RNF168, restricts end resection	[17,40,65,113,114,115,116,117,118,119,120]
DUB3/USP17L2	USP		H2A(X)	Antagonizes RNF8/RNF168, promotes HR	[33,107,121]
MYSM1/2A-DUB	JAMM		H2AK119ub1	Promotes HR ^#^, transcriptional regulation ^#^, chromatin accessibility ^#^	[109,110,122,123,124,125,126]
POH1/PSMD14	JAMM	26S proteasomal subunits	H2A K63-linked chains	Regulates 53BP1, promotes HR	[109,117,127,128,129,130,131]
USP3	USP		H2A(X)K13/15 H2AX119ub1	Protection from spontaneous DNA damage, antagonizes RNF168	[18,109,110,132,133,134,135,136,137]
USP11	USP		γH2AX H2AK119ub1 H2BK120ub1	Promotes HR, transcriptional regulation ^#^, chromatin accessibility ^#^	[33,109,137,138]
USP16/Ubp-M	USP		H2AK119ub1 H2AK13/15 *	Transcriptional regulation, antagonizes RNF8/RNF168 ^#^	[65,110,136,139,140,141,142,143]
USP22	USP	ATXN7, ATXN7L3, ENY2	H2BK120ub1	Promotes NHEJ, transcriptional regulation ^#^	[91,110,144,145,146,147,148,149,150]
USP26, USP29, USP37	USP		H2AK13/15 K63-linked chains * H2BK120ub1 *	Promotes HR, antagonizes RNF8/RNF168, antagonizes BRCA1-A complex formation	[110,151,152,153,154,155]
USP44	USP	TBL1X, TBL1XR1, HDAC3, NCOR1, NCOR2 (NCoR complex)	H2AK13/15 γH2AX H2BK120ub1 *	Counteracts RNF8/RNF168 to regulate IRIF assembly	[109,110,135,156,157]
USP48	USP		H2AK125/127/129	Counteracts BRCA1-BARD1 to restrict 53BP1 repositioning and end resection	[158]
USP51	USP	ATXN7L3, ENY2	H2AK13/15 H2AK119ub1 * K27-linked chains * H2BK120ub1 *	Termination of DSB repair, transcriptional regulation ^#^, chromatin accessibility ^#^	[136,146,159,160]

* Designates a known histone substrate of the indicated DUB that has yet to be implicated in the DSB response; # Designates a potential function of the indicated DUB given its known histone substrates.

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
