# Peer review of "Regulation of Histone Ubiquitination in Response to DNA Double Strand Breaks"

_cells, 2020, doi:10.3390/cells9071699_

Round 1

Reviewer 1 Report

Comments to the Author
In this study the author(s) have attempted to provide a summary of the current understanding of histone ubiquitination and deubiquintination events in response to DNA double strand breaks (DSBs). In a broad sense the manuscript can be divided into two segments where the former deals with the current understanding of how histone ubiquitination contributes to successful repair of DSBs. The later segments discuss the DUBs that remove these histone ubiquitination and regulate the DSBs repair. The authors provided an up-to-date account on how DSBs responses are driven by ubiquitination.  

I have a few comments that might help improve the manuscript -

  1. In addition to histone ubiquitination, other histone modifications, such as phosphorylation, acetylation, methylation, also play important roles in the repair of DSBs. The author(s) focused on histone ubiquitination in this review. However, the crosstalk between histone ubiquitination and other histone modifications in DDR needs to be elaborated and discussed.
  2. Currently two models of the recruitment of RNF168 to DSBs were proposed. Both histone H1 and L3MTBL2 are reported to recruit RNF168. Accordingly, the authors need to revise the fig.1A.
  3. A typo of 53BP1 in Fig.1A needs to be corrected.

Author Response

Point by point responses to Reviewer #1

1) In addition to histone ubiquitination, other histone modifications, such as phosphorylation, acetylation, methylation, also play important roles in the repair of DSBs. The author(s) focused on histone ubiquitination in this review. However, the crosstalk between histone ubiquitination and other histone modifications in DDR needs to be elaborated and discussed.

Response: Lines 703-721 within the new Section 4 have been added detailing examples of histone ubiquitination crosstalk with other PTMs including phosphorylation, acetylation, methylation, PARylation, and SUMOylation. Readers are additionally referred to two recent reviews that expand on this topic.

2) Currently, two models of the recruitment of RNF168 to DSBs were proposed. Both histone H1 and L3MTBL2 are reported to recruit RNF168. Accordingly, the authors need to revise the fig.1A.

Response: Figure 1A has been modified to include L3MBTL2-mediated recruitment of RNF168 and the corresponding caption has been adjusted to reflect this change (line 198). L3MBTL2 has also been discussed in lines 94-96 of the text.  

3) A typo of 53BP1 in Fig.1A needs to be corrected.

Response: The typo of “51BP1” has been corrected to “53BP1”.

Reviewer 2 Report

This review article is about the regulation of histone ubiquitination in response to DNA double strand breaks (DSBs), with a particular focus on deubiquitinated enzymes (DUBs). The authors discussed the ubiquitin signalling and its role in the recruitment of DNA repair proteins (RNF8-RNF168 axis) and transcriptional repression (H2AK119ub) at the site of DSBs. Further, they discussed about the non-canonical H2BK120ub at DNA damage site and the potential contribution of H3 and H4 ubiquitination for DSB repair. Finally, the authors listed the known DUBs in DSB repair and explained their associated protein complexes, histone substrate targets and significance for DSB repair.

This manuscript is nicely written and gives a comprehensive overview of DUBs in DSB repair. However, before this manuscript could be accepted for publication, it should be improved at several points.

Specifically, the authors have to include and discuss the following points:

  1. Ataxin 3 is a well-established deubiquitinating enzyme, recently shown to be involved in DSB repair with VCP/p97. It regulates the histone ubiquitination at DSBs through the regulation of RNF8, the first E3 ubiquitin ligase at the sites of DSBs, and Ataxin 3 is essential for efficient NHEJ repair pathway. Thus, Ataxin 3 should be included and discussed in this manuscript (Singh et al. EMBO J 2019).
  2. VCP/p97 is an ATPase known to regulate both K48 and K63 ubiquitin signals by segregating ubiquitinated substrates at DSBs. VCP/p97 need to be discussed in detail as it is a major modulator of ubiquitin signal and essential for both NHEJ and HR repair pathways (e.g. Meerang et al. NCB, 2011, Torrecilla et al. Philos Trans R Soc Lond B Biol Sci, 2017).
  3. USP7 is a DUB that can alter the ubiquitination status of both H2A (Luo et al., 2015) and H2B (Knaap et al., 2005) to regulate the gene expression. It has been found to be recruited at the sites of DSBs and involved in DSB repair by stabilising RNF168 (Zhu et al., 2015) and histone demethylase PHF8 (Wang et al., 2016).
  4. USP10, a cytoplasmic ubiquitin-specific protease, can deubiquitinate p53 to promote its stability, nuclear localisation and activation of apoptosis in response to DSBs (Yuan et al., 2010). USP10 can deubiquitinate H2A.Z to regulate the androgen receptor mediated gene expression (Draker et al., 2011).
  5. USP21 deubiquitinates BRCA2 to promote its stability and efficient Rad51 loading to control DNA repair (HR) and tumour growth (Liu et at., 2017). It has also been found to remove inhibitory Ub from H2A to promote transcriptional activation (Nakagawa et al., 2008).
  6. Discussing the clinical relevance of DUBs in DSB repair and providing the recent updates of targeting DUBs for therapeutics will further make this manuscript more attractive for readers. 

Author Response

Point by point responses to reviewer #2

1) Ataxin 3 is a well-established deubiquitinating enzyme, recently shown to be involved in DSB repair with VCP/p97. It regulates the histone ubiquitination at DSBs through the regulation of RNF8, the first E3 ubiquitin ligase at the sites of DSBs, and Ataxin 3 is essential for efficient NHEJ repair pathway. Thus, Ataxin 3 should be included and discussed in this manuscript (Singh et al. EMBO J 2019).

Response: Work from Singh et al implicating Ataxin 3 along with p97 in the regulation of RNF8 homeostasis in order to prevent RNF8 hyperaccumulation at DSB sites, thereby enabling efficient NHEJ repair has been discussed in lines 98-117.

2) VCP/p97 is an ATPase known to regulate both K48 and K63 ubiquitin signals by segregating ubiquitinated substrates at DSBs. VCP/p97 need to be discussed in detail as it is a major modulator of ubiquitin signal and essential for both NHEJ and HR repair pathways (e.g. Meerang et al. NCB, 2011, Torrecilla et al. Philos Trans R Soc Lond B Biol Sci, 2017).

Response: The segregase activity of p97 towards ubiquitinated substrates and the importance of this activity for proper DSB repair has been discussed in lines 98-117. p97 is discussed further in lines 142-150 for its involvement in L3MBTL1 extraction from chromatin and 53BP1 recruitment to H4K20me2.

3) USP7 is a DUB that can alter the ubiquitination status of both H2A (Luo et al., 2015) and H2B (Knaap et al., 2005) to regulate the gene expression. It has been found to be recruited at the sites of DSBs and involved in DSB repair by stabilising RNF168 (Zhu et al., 2015) and histone demethylase PHF8 (Wang et al., 2016).

Response: The ability for USP7 to modify H2A and H2B as well as its role in deubiquitinating and stabilizing RNF168 and PHF8 have been discussed in lines 691-697.

4) USP10, a cytoplasmic ubiquitin-specific protease, can deubiquitinate p53 to promote its stability, nuclear localisation and activation of apoptosis in response to DSBs (Yuan et al., 2010). USP10 can deubiquitinate H2A.Z to regulate the androgen receptor mediated gene expression (Draker et al., 2011).

Response: USP10-mediated regulation of gene expression through histone deubiquitination has been referenced in lines 75-76. The deubiquitination and stabilization of p53 by USP10 to promote nuclear localization and apoptosis upon DSB induction has been discussed in lines 699-701.

5) USP21 deubiquitinates BRCA2 to promote its stability and efficient Rad51 loading to control DNA repair (HR) and tumour growth (Liu et at., 2017). It has also been found to remove inhibitory Ub from H2A to promote transcriptional activation (Nakagawa et al., 2008).

Response: USP21 regulation of transcription through H2A deubiquitination was referenced in lines 75-76. The ability for USP21 to deubiquitinate and stabilize BRCA2 as a means of regulating HR has been discussed in lines 698-699.

6) Discussing the clinical relevance of DUBs in DSB repair and providing the recent updates of targeting DUBs for therapeutics will further make this manuscript more attractive for readers. 

Response: Section 5 (lines 723-750) has been added to address this point.

Reviewer 3 Report

This review by Aquila and Atanassov deals with the functions of histone ubiquitination during the response to DNA double strand breaks. This is a very important topic in which significant advances have been made in the last years. In addition, the manuscript pays special attention to the regulation of double strand break repair by the deubiquitinase enzymes that limit the modification of histones around the break. This review is quite thorough and offers a good resource to understand the functions of histone ubiquitination in this process. It is well written and, in general, well explained although there are some important aspects of the topic to be included and discussed. Since these aspects are quite recent and very relevant to the process of DNA double strand break repair they need to be included and discussed in the manuscript before it is considered for publication.

The major points that the authors should include/change are:

  1. There is no mention to the ubiquitination of histone H2A at K125/127/129. Recent work by the Morris lab and others has revealed that BRCA1/BARD1 ubiquitinates H2A at this residue to control resection of DNA at the break and to promote homologous recombination. Further, they have identified USP48 as a deubiquitinase that counteracts the action of BRCA1/BARD1. This work is key in the field since it offers a new vision where the ubiquitination of different lysine residues in H2A participates in different steps of the repair process. Along these lines, the authors should discuss the functions of the ubiquitination of K13/15 (recruitment of 53BP1, BRCA1-A complex) and K125/127/129 (resection) in the steps of pathway choice and control of resection, separate from the functions of H2A K119 ubiquitination that is mainly working on the control of transcription. Further, since BRCA1 is part of the BRCA1-A complex and also works as a dimer with BARD1 the authors should discuss any potential inter-relationships between them. Is the amount of BRCA1 limited? Does the formation of these complexes depend on the amount of BRCA1 available? A more detailed explanation of the steps of repair and pathway choice will be needed to address this point
  2. The recruitment of RAP80 has been shown to depend on SUMO in addition to ubiquitin. It has been postulated that the SUMO dependent ubiquitin ligase RNF4 is responsible for this dependency. The interplay between SUMO and ubiquitin has been shown to be increasingly relevant in chromatin and the authors should include the role of SUMO and RNF4 in the recruitment of RAP80.
  3. The ubiquitination of histones H3 and H4 is poorly understood and its relevance to the repair of DNA double strand breaks has not been clearly established. Further, the residues that are modified and the deubiquitinases that regulate this process are still unknown. In consequence, the authors could reduce the extension and importance given to these modifications, since there is still much work to be done before their functions are determined
  4. The section regarding deubiquitinases is a bit confusing since it is organized as a list with no particular order. In order to help understand the different processes regulated by these enzymes it could be helpful to group them by the functions they perform: regulation of K63 chain formation, modification of H2AK13/15, H2AK119, H2AK125/127/129, H2BK120 and their consequences for chromatin status (transcription inhibition, access), the recruitment of different proteins, pathway choice and DNA resection. Since many of these deubiquitinases play overlapping roles this organization would help to better understand their functions.

In addition, there are other minor points that should be taken care of:

  1. The definition of the different modifications by the ubiquitin system needs to be better explained: mono-ubiquitination, poly-mono-ubiquitination, chain formation and branched/mixed chain formation. It is important to understand that mono-ubiquitination on several residues in the same histone can take place.
  2. The nomenclature of the lysine residues within the manuscript is not constant. The authors should review the numbering of the lysine to make it consistent.
  3. The access of 53BP1 to chromatin is regulated by the binding to H4K20me2. The authors refer to the functions of JMJD2A in this context, they should include a similar role found for L3MBTL1 in blocking the access to H4K20me2.

Author Response

Point by point responses to reviewer #3

Major points:

1) There is no mention to the ubiquitination of histone H2A at K125/127/129. Recent work by the Morris lab and others has revealed that BRCA1/BARD1 ubiquitinates H2A at this residue to control resection of DNA at the break and to promote homologous recombination. Further, they have identified USP48 as a deubiquitinase that counteracts the action of BRCA1/BARD1. This work is key in the field since it offers a new vision where the ubiquitination of different lysine residues in H2A participates in different steps of the repair process. Along these lines, the authors should discuss the functions of the ubiquitination of K13/15 (recruitment of 53BP1, BRCA1-A complex) and K125/127/129 (resection) in the steps of pathway choice and control of resection, separate from the functions of H2A K119 ubiquitination that is mainly working on the control of transcription. Further, since BRCA1 is part of the BRCA1-A complex and also works as a dimer with BARD1 the authors should discuss any potential inter-relationships between them. Is the amount of BRCA1 limited? Does the formation of these complexes depend on the amount of BRCA1 available? A more detailed explanation of the steps of repair and pathway choice will be needed to address this point

Response: Section 2.3 (lines 190-195; 216-259) has been added to discuss work by the Morris lab demonstrating that BRCA1-BARD1 mediated ubiquitination of H2A at K125/127/129 impacts DSB repair pathway decision through SMARCAD1 activity. Removal of this mark by USP48 has been discussed in the new Section 3.12 (lines 627-651). Ubiquitination events at H2AK125/127/129 have been depicted in the new Figure 1C and referenced in the corresponding figure caption (lines 206-210). Distinct BRCA1 complexes (BRCA1-A, BRCA1-B, BRCA1-C, BRCA1-BARD1) and the inter-relationships between them have been discussed in Section 2.3 (lines 245-259).

2) The recruitment of RAP80 has been shown to depend on SUMO in addition to ubiquitin. It has been postulated that the SUMO dependent ubiquitin ligase RNF4 is responsible for this dependency. The interplay between SUMO and ubiquitin has been shown to be increasingly relevant in chromatin and the authors should include the role of SUMO and RNF4 in the recruitment of RAP80.

Response: The ability for RAP80 to recognize SUMO-ubiquitin hybrid chains catalyzed by RNF4 has been added in lines 125-133.

3) The ubiquitination of histones H3 and H4 is poorly understood and its relevance to the repair of DNA double-strand breaks has not been clearly established. Further, the residues that are modified and the deubiquitinases that regulate this process are still unknown. In consequence, the authors could reduce the extension and importance given to these modifications, since there is still much work to be done before their functions are determined

Response: We still wish to highlight the potential implications of H3 and H4 ubiquitination events in DSB repair in Section 2.5. However, the old Figure 1D illustrating these modifications has been removed and the corresponding figure caption (lines 212-214) has been shortened to reduce the importance given to these modifications.

4) The section regarding deubiquitinases is a bit confusing since it is organized as a list with no particular order. In order to help understand the different processes regulated by these enzymes it could be helpful to group them by the functions they perform: regulation of K63 chain formation, modification of H2AK13/15, H2AK119, H2AK125/127/129, H2BK120 and their consequences for chromatin status (transcription inhibition, access), the recruitment of different proteins, pathway choice and DNA resection. Since many of these deubiquitinases play overlapping roles this organization would help to better understand their functions.

Response: Several DUBs mentioned in this review have multiple histone substrates and for others, the specific lysine residues they modify have yet to be defined. This has made it difficult to organize Section 3 based on function or substrate. Instead, DUBs have been listed in alphabetical order. We hope the reviewer will accept this approach.

Minor points:

1) The definition of the different modifications by the ubiquitin system needs to be better explained: mono-ubiquitination, poly-mono-ubiquitination, chain formation and branched/mixed chain formation. It is important to understand that mono-ubiquitination on several residues in the same histone can take place.

Response: The ubiquitination definitions found in lines 51-58 have been expanded to include monoubiquitination and multi-monoubiquitination as well as linear, mixed, and branched polyubiquitin chains.

2) The nomenclature of the lysine residues within the manuscript is not constant. The authors should review the numbering of the lysine to make it consistent.

Response: Throughout the text, monoubiquitination has been denoted by “ub1” following the histone and lysine residue (i.e. H2AK119ub1). Polyubiquitin chains have now been clearly specified i.e. K63-linked polyubiquitin chains.

3) The access of 53BP1 to chromatin is regulated by the binding to H4K20me2. The authors refer to the functions of JMJD2A in this context, they should include a similar role found for L3MBTL1 in blocking the access to H4K20me2.

Response: The polyubiquitination of L3MBTL1 by RNF8 and subsequent proteasomal degradation in order to facilitate 53BP1 binding to H4K20me2 has been discussed in lines 139-150.

Round 2

Reviewer 3 Report

The new version of the revision has included all the changes that I had suggested and addressed all my comments. Thanks to the authors for their nice and comprehensive work. I think that the paper is now ready for its publication.